# LEARNING TO BALANCE:
# BAYESIAN META-LEARNING FOR IMBALANCED AND OUT-OF-DISTRIBUTION TASKS

**Hae Beom Lee**[1*]**, Hayeon Lee**[1*]**, Donghyun Na**[2*]**,**
**Saehoon Kim**[3]**, Minseop Park**[3]**, Eunho Yang**[1,3]**, Sung Ju Hwang**[1,3]
KAIST[1], TmaxData[2], AITRICS[3], South Korea
{haebeom.lee, hayeon926, eunhoy, sjhwang82}@kaist.ac.kr
donghyun_na@tmax.co.kr, {shkim, mike_seop}@aitrics.com

## ABSTRACT

While tasks could come with varying the number of instances and classes in realistic settings, the existing meta-learning approaches for few-shot classification assume that the number of instances per task and class is fixed. Due to such restriction, they learn to *equally* utilize the meta-knowledge across all the tasks, even when the number of instances per task and class largely varies. Moreover, they do not consider distributional difference in unseen tasks, on which the meta-knowledge may have less usefulness depending on the task relatedness. To overcome these limitations, we propose a novel meta-learning model that *adaptively balances* the effect of the meta-learning and task-specific learning within each task. Through the learning of the balancing variables, we can decide whether to obtain a solution by relying on the meta-knowledge or task-specific learning. We formulate this objective into a Bayesian inference framework and tackle it using variational inference. We validate our Bayesian Task-Adaptive Meta-Learning (Bayesian TAML) on multiple realistic task- and class-imbalanced datasets, on which it significantly outperforms existing meta-learning approaches. Further ablation study confirms the effectiveness of each balancing component and the Bayesian learning framework.

## 1 INTRODUCTION

Despite the success of deep learning in many real-world tasks such as visual recognition and machine translation, such good performances are achievable at the availability of large training data, and many fail to generalize well in small data regimes. To overcome this limitation of conventional deep learning, recently, researchers have explored meta-learning (Schmidhuber, 1987; Thrun & Pratt, 1998) approaches, whose goal is to learn a model that generalizes well over distribution of tasks, rather than instances from a single task, in order to utilize the obtained meta-knowledge across tasks to compensate for the lack of training data for each task.

However, so far, most existing meta-learning approaches (Santoro et al., 2016; Vinyals et al., 2016; Snell et al., 2017; Ravi & Larochelle, 2017; Finn et al., 2017; Li et al., 2017) have only targeted an artificial scenario where all tasks participating in the multi-class classification problem have equal number of training instances per class. Yet, this is a highly restrictive setting, as in real-world scenarios, tasks that arrive at the model may have different training instances (task imbalance), and within each task, the number of training instances per class may largely vary (class imbalance). Moreover, the new task may come from a distribution that is different from the task distribution the model has been trained on (out-of-distribution task) (See (a) of Figure 1).

Under such a realistic setting, the meta-knowledge may have a varying degree of utility to each task. Tasks with small number of training data, or close to the tasks trained in meta-training step may want to rely mostly on meta-knowledge obtained over other tasks, whereas tasks that are out-of-distribution or come with more number of training data may obtain better solutions when trained in a task-specific

---

[*]Equal contribution

Figure 1: **Concept.** (a) To handle task imbalance, class imbalance and out-of-distribution (OOD) tasks, we introduce task-specific balancing variables $\gamma^\tau$, $\omega^\tau$, and $\mathbf{z}^\tau$. (b) With those variables, we learn to balance between the meta-knowledge $\theta$ and task-specific update to handle imbalances and distributional shift.

manner. Furthermore, for multi-class classification, we may want to treat the learning for each class differently to handle class imbalance. Thus, to optimally leverage meta-learning under various imbalances, it would be beneficial for the model to task- and class-adaptively decide how much to use from the meta-learner, and how much to learn specifically for each task and class.

To this end, we propose a novel Bayesian meta-learning framework, which we refer to as Bayesian Task-Adaptive Meta-Learning (Bayesian TAML), that learns variables to adaptively balance the effect of meta- and task-specific learning. Specifically, we first obtain set-representations for each task, which are learned to convey useful statistics about the task or class distribution, such as mean, variance, and cardinality (the number of elements in the set), and then learn the distribution of three balancing variables a function of the set: 1) *task-dependent learning rate multiplier*, which decides how far away to deviate from the meta-knowledge, when performing task-specific learning. Tasks with higher shots could benefit from taking gradient steps afar, while tasks with few shots may need to stay close to the initial parameter. 2) *class-dependent learning rate*, which decides how much information to use from each class, to automatically handle class imbalance where the number of instances per class can largely vary. 3) *task-dependent modulator for initial model parameter*, which modifies the shared initialization for each task, such that each task can decide how much and what to use from the shared initial model parameter and what to ignore based on its set representation. This is especially useful when handling out-of-distribution task, which may need to ignore some of the meta-knowledge.

We validate our model on CIFAR-FS and miniImageNet dataset, as well as a new dataset that consists of heterogeneous datasets, under a scenario where every class in each episode can have *any* number of shots, that leads to task and class imbalance, and where the dataset at meta-test time is different from that of meta-training time. The experimental results show that our Bayesian TAML significantly improves the performance over the existing approaches under these realistic scenarios. Further analysis of each component reveals that the improvement is due to the effectiveness of the balancing terms for handling task and class imbalance, and out-of-distribution tasks.

To summarize, our contribution in this work is threefold:

- We consider a novel problem of meta-learning under a realistic task distribution, where the number of instances across classes and tasks could largely vary, or the unseen task at the meta-test time is largely different from the seen tasks.

- For effective meta-learning with such imbalances, we propose a Bayesian task-adaptive meta-learning (Bayesian TAML) framework that can adaptively adjust the effect of the meta-learner and the task-specific learner, differently for each task and class.

- We validate our model on realistic imbalanced few-shot classification tasks with a varying number of shots per task and class and show that it significantly outperforms existing meta-learning models.

## 2 RELATED WORK

**Meta-learning** Meta-learning (Schmidhuber, 1987; Thrun & Pratt, 1998) is an approach to learn a model to generalize over a distribution of task. The approaches in general can be categorized into either memory-based, metric-based, and optimization-based methods. A memory-based approach (Santoro et al., 2016) learns to store correct instance and label into the same memory slot and retrieve it later, in a task-generic manner. Metric-based approaches learn a shared metric

space (Vinyals et al., 2016; Snell et al., 2017). Snell et al. (2017) defines the distance between the instance and the class prototype, such that the instances are closer to their correct prototypes than to others. As for optimization-based meta-learning, MAML (Finn et al., 2017) learns a shared initialization parameter that is optimal for any tasks within few gradient steps from the initial parameter. Meta-SGD (Li et al., 2017) improves upon MAML by learning the learning rate differently for each parameter. For effective learning of a meta-learner, meta-learning approaches adopt the episodic training strategy (Vinyals et al., 2016) which trains and evaluates a model over a large number of tasks, which are called meta-training and meta-test phase, respectively. However, existing approaches only consider an artificial scenario which samples the classification of classes with exactly the same number of training instances, both within each episode and across episodes. On the other hand, we consider a more challenging scenario where the number of shots per class and task could vary at each episode, and that the task given at meta-test time could be an out-of-distribution task.

**Task-adaptive meta-learning**   The goal of learning a single meta-learner that works well for all tasks may be overly ambitious and leads to suboptimal performances for each task. Thus recent approaches adopt task-adaptively modified meta-learning models. Oreshkin et al. (2018) proposed to learn the temperature scaling parameter to work with the optimal similarity metric. Qiao et al. (2018) also suggested a model that generates task-specific parameters for the network layers, but it only trains with many-shot classes, and implicitly expects generalization to few-shot cases. Rusu et al. (2019) proposed a network type task-specific parameter producer, and Lee & Choi (2018) proposed to differentiate the network weights into task-shared and task-specific weights. Our model also aims to obtain task-specific parameter for each task, but is rather focused on learning how to balance between the meta-learning and task-/class-specific learning.

**Probabilistic meta-learning**   Recently, a probabilistic version of MAML has been proposed (Finn et al., 2018), where they interpret a task-specific gradient update as a posterior inference process under variational inference framework. Kim et al. (2018) proposed Bayesian MAML with a similar motivation but with a stein variational inference framework and chaser loss. Gordon et al. (2019) proposed a probabilistic meta-learning framework where the parameter for a novel task is rapidly estimated under decision theoretic framework, given a set representation of a task. The motivation behind these works is to represent the inherent uncertainty in few-shot classification tasks. Our model also uses Bayesian modeling, but it focuses on leveraging the uncertainties of the meta-learner and the gradient-direction in order to balance between meta- and task- or class-specific learning.

## 3   LEARNING TO BALANCE

We first introduce notations and briefly recap the model-agnostic meta-learning (MAML) by Finn et al. (2017). Suppose a task distribution $p(\tau)$ that randomly generates task $\tau$ consisting of a training set $\mathcal{D}^\tau = \{\mathbf{X}^\tau, \mathbf{Y}^\tau\}$ and a test set $\tilde{\mathcal{D}}^\tau = \{\tilde{\mathbf{X}}^\tau, \tilde{\mathbf{Y}}^\tau\}$. Then, the goal of MAML is to meta-learn the initial model parameter $\boldsymbol{\theta}$ as a meta-knowledge to generalize over the task distribution $p(\tau)$, such that we can easily obtain the task-specific predictor $\boldsymbol{\theta}^\tau$ in a single (or a few) gradient step from the initial $\boldsymbol{\theta}$. Toward this goal, MAML optimizes the following gradient-based meta-learning objective:

$$\min_{\boldsymbol{\theta}} \sum_{\tau \sim p(\tau)} \mathcal{L}(\boldsymbol{\theta} - \alpha \nabla_{\boldsymbol{\theta}} \mathcal{L}(\boldsymbol{\theta}; \mathcal{D}^\tau); \tilde{\mathcal{D}}^\tau) \qquad (1)$$

where $\alpha$ denotes stepsize and $\mathcal{L}$ denotes empirical loss such as negative log-likelihood of observations. Note that by meta-learning the initial point $\boldsymbol{\theta}$, the task-specific predictor $\boldsymbol{\theta}^\tau = \boldsymbol{\theta} - \alpha \nabla_{\boldsymbol{\theta}} \mathcal{L}(\boldsymbol{\theta}; \mathcal{D}^\tau)$ can minimize the test loss $\mathcal{L}(\cdot; \tilde{\mathcal{D}}^\tau)$ even with $\mathcal{D}^\tau$ consisting of only a few samples. We can easily extend the Eq. (1), such that we obtain $\boldsymbol{\theta}^\tau$ with more than one inner-gradient steps from the initial $\boldsymbol{\theta}$.

However, the existing MAML framework has the following limitations that prevent the model from efficiently solving real-world problems involving class/task imbalance and out-of-distribution tasks.

1. **Class imbalance.** MAML does not provide any framework to handle class imbalance within each task. Therefore, classes with large number of training instances (head classes) may dominate the task-specific learning during the inner-gradient steps, yielding low performance on classes with fewer shots (tail classes).

2. **Task imbalance.** The model has a fixed number of inner-gradient steps and stepsize $\alpha$ across all tasks, which prevents the model from adaptively deciding how much to resort to

the meta-knowledge or how much to learn from the given dataset, depending on the number of the training examples per task.

3. **Out-of-distribution tasks.** The model assumes that the initial model parameter $\boldsymbol{\theta}$ will be equally useful for the unseen tasks, but for unseen tasks that are significantly different from the previously seen tasks, the initial parameter may be less useful.

## 3.1 TASK-ADAPTIVE META-LEARNING (TAML)

As shown in Figure 1 for the concepts, we introduce three balancing variables $\boldsymbol{\omega}^\tau, \boldsymbol{\gamma}^\tau, \mathbf{z}^\tau$ to tackle each problem mentioned above. How to generate these variables will be described in Section 4. Also, see the experimental section for how to generate the realistic tasks with class and task imbalance.

**Tackling class imbalance.** To handle class imbalance, we vary the learning rate of class-specific gradient for each inner-optimization step. Specifically, for class $c = 1, \ldots, C$, we introduce a set of class-specific scalars $\boldsymbol{\omega}^\tau = (\omega_1^\tau, \ldots, \omega_C^\tau) \in [0, 1]^C$, which are multiplied to each of the class-specific gradients $\nabla_{\boldsymbol{\theta}} \mathcal{L}(\boldsymbol{\theta}; \mathcal{D}_1^\tau), \ldots, \nabla_{\boldsymbol{\theta}} \mathcal{L}(\boldsymbol{\theta}; \mathcal{D}_C^\tau)$, where $\mathcal{D}_c^\tau$ is the set of instances and labels for class $c$. We expect $\omega_c^\tau$ to be large for tail-classes with small number of training instances, such that they could be considered more in the inner-optimization steps. We generate $\boldsymbol{\omega}^\tau$ with Softmax function, and denote the input to the function as $\tilde{\boldsymbol{\omega}}^\tau$.

**Tackling task imbalance.** To control whether the model parameter for the current task stays close to the initial parameter or deviate far from it, we introduce task-dependent learning-rate multipliers $\boldsymbol{\gamma}^\tau = (\gamma_1^\tau, \ldots, \gamma_L^\tau) \in [0, \infty)^L$, such that for each layer $l = 1, \ldots, L$, the learning rate becomes $\gamma_1^\tau \alpha, \gamma_2^\tau \alpha, \ldots, \gamma_L^\tau \alpha$[1]. We expect $\boldsymbol{\gamma}^\tau$ to be large for large tasks, such that they rely more on task-specific updates, while small tasks use small $\boldsymbol{\gamma}^\tau$ to benefit more from the meta-knowledge. To amplify the step-size difference between large and small tasks, we generate $\boldsymbol{\gamma}^\tau$ with an exponential function, and denote the input to the function as $\tilde{\boldsymbol{\gamma}}^\tau$.

**Tackling out-of-distribution tasks.** Finally, we introduce $\mathbf{z}^\tau$ which modulates the initial parameter $\boldsymbol{\theta}$ for each task. We expect $\mathbf{z}^\tau$ to learn to relocate the initial $\boldsymbol{\theta}$ to a new starting point, such that out-of-distribution (OOD) tasks can deviate much from the shared initialization $\boldsymbol{\theta}$ if the current initialization is suboptimal for the given task. Specifically, we use $\mathbf{z}^\tau = \mathbf{1} + \tilde{\mathbf{z}}^\tau$ for the channel of the convolutional network weights and $\mathbf{z}^\tau = \tilde{\mathbf{z}}^\tau$ for the biases, which modify the initial parameter as follows: $\boldsymbol{\theta}_0 \leftarrow \boldsymbol{\theta} \circ \mathbf{z}^\tau$ for the weights and $\boldsymbol{\theta}_0 \leftarrow \boldsymbol{\theta} + \mathbf{z}^\tau$ for the biases, where $\boldsymbol{\theta}_0$ denotes the new initialization modulated by $\mathbf{z}^\tau$. We denote this operation as $\boldsymbol{\theta} * \mathbf{z}^\tau$, which is similar to task-dependent modulation of batch normalization parameters (Oreshkin et al., 2018; Requeima et al., 2019).

**A unified framework.** Finally, we assemble all these components together into a single unified framework. With a slight abuse of notations, the update rule can be written as follows:

$$\boldsymbol{\theta}_0 = \boldsymbol{\theta} * \mathbf{z}^\tau, \tag{2}$$

$$\boldsymbol{\theta}_k = \boldsymbol{\theta}_{k-1} - \boldsymbol{\gamma}^\tau \circ \boldsymbol{\alpha} \circ \sum_{c=1}^{C} \omega_c^\tau \nabla_{\boldsymbol{\theta}_{k-1}} \mathcal{L}(\boldsymbol{\theta}_{k-1}; \mathcal{D}_c^\tau) \quad \text{for} \ k = 1, \ldots, K \tag{3}$$

where $\boldsymbol{\alpha}$ is a multi-dimensional global learning rate vector that is learned (Li et al., 2017), and the multiplication operator $\circ$ is appropriately defined. The last step $\boldsymbol{\theta}_K$ corresponds to the task-specific predictor $\boldsymbol{\theta}^\tau$.

## 3.2 BAYESIAN TASK-ADAPTIVE META-LEARNING

We now introduce the variational inference framework for the *input* of the three balancing variables, that we previously denote as $\tilde{\boldsymbol{\omega}}^\tau, \tilde{\boldsymbol{\gamma}}^\tau, \tilde{\mathbf{z}}^\tau$. Bayesian modeling allows to incorporate randomness in the posterior of those variables. In MAML framework, such randomness generates the ensemble of diverse task-specific predictors, which allows to effectively exploit the information in the given dataset $\mathcal{D}^\tau$. Bayesian modeling also allows to find more robust latent structures of those balancing variables (See Figure 7).

---

[1]We initially exponentially decayed the learning rate $\gamma \in [0, 1]$, but found it suboptimal as it prevents the learning rate from growing for many-shot tasks.

Firstly, define $\mathbf{X}^\tau = \{\mathbf{x}_n^\tau\}_{n=1}^{N_\tau}$ and $\mathbf{Y}^\tau = \{\mathbf{y}_n^\tau\}_{n=1}^{N_\tau}$ for training, and $\tilde{\mathbf{X}}^\tau = \{\tilde{\mathbf{x}}_m^\tau\}_{m=1}^{M_\tau}$ and $\tilde{\mathbf{Y}}^\tau = \{\tilde{\mathbf{y}}_m^\tau\}_{m=1}^{M_\tau}$ for test. Let $\phi^\tau$ denote the collection of $\tilde{\omega}^\tau$, $\tilde{\gamma}^\tau$ and $\tilde{\mathbf{z}}^\tau$ for uncluttered notation. Then, inspired by Gordon et al. (2019) and Finn et al. (2018), we define the generative process for meta-learning framework as follows for each task $\tau$ (See Figure 2):

$$p(\mathbf{Y}^\tau, \tilde{\mathbf{Y}}^\tau, \phi^\tau | \mathbf{X}^\tau, \tilde{\mathbf{X}}^\tau; \boldsymbol{\theta}) = p(\phi^\tau) \prod_{n=1}^{N_\tau} p(\mathbf{y}_n^\tau | \mathbf{x}_n^\tau, \phi^\tau; \boldsymbol{\theta}) \prod_{m=1}^{M_\tau} p(\tilde{\mathbf{y}}_m^\tau | \tilde{\mathbf{x}}_m^\tau, \phi^\tau; \boldsymbol{\theta}) \qquad (4)$$

for the complete data likelihood. Note that the deterministic $\boldsymbol{\theta}$ is shared across all the tasks.

## 4 VARIATIONAL INFERENCE

The goal of learning for each task $\tau$ is to maximize the conditional log-likelihood of the joint dataset $\tilde{\mathcal{D}}^\tau$ and $\mathcal{D}^\tau$: $\log p(\tilde{\mathbf{Y}}^\tau, \mathbf{Y}^\tau | \tilde{\mathbf{X}}^\tau, \mathbf{X}^\tau; \boldsymbol{\theta})$. However, solving it involves the true posterior $p(\phi^\tau | \mathcal{D}^\tau, \tilde{\mathcal{D}}^\tau)$, which is intractable. Thus, we resort to amortized variational inference with a tractable form of approximate posterior $q(\phi^\tau | \mathcal{D}^\tau, \tilde{\mathcal{D}}^\tau; \boldsymbol{\psi})$ parameterized by $\boldsymbol{\psi}$. We let the three variables share the same inference network pipeline to minimize the computational cost. Further, similarly to Ravi & Beatson (2019), we drop the dependency on the test dataset $\tilde{\mathcal{D}}^\tau$ for the approximate posterior, in order to make the two different pipelines consistent; one for meta-training where we observe the whole test dataset, and the other for meta-testing where the test labels are unknown. The form of our approximate posterior is now $q(\phi^\tau | \mathcal{D}^\tau; \boldsymbol{\psi})$. It greatly simplifies the inference framework, while ensuring to have a valid lower bound of the log evidence. Also, considering that performing the inner-gradient steps with

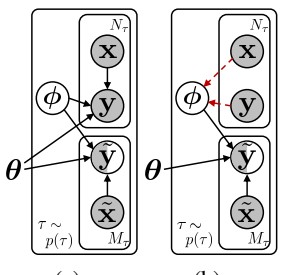

Figure 2: **Graphical model.** (a) Generative process. (b) Inference.

the training dataset $\mathcal{D}^\tau$ automatically maximizes the training log-likelihood in MAML framework, we slightly modify the objective so that the expected log-likelihood term only involves the test examples with the appropriate scaling factor. The resultant form of the approximated lower bound that suits for our meta-learning purpose is as follows:

$$L_{\boldsymbol{\theta}, \boldsymbol{\psi}}^\tau = \frac{N_\tau + M_\tau}{M_\tau} \sum_{m=1}^{M_\tau} \mathbb{E}_{q(\phi^\tau | \mathcal{D}^\tau; \boldsymbol{\psi})} \Big[ \log p(\tilde{\mathbf{y}}_m^\tau | \tilde{\mathbf{x}}_m^\tau, \phi^\tau; \boldsymbol{\theta}) \Big] - \mathrm{KL}[q(\phi^\tau | \mathcal{D}^\tau; \boldsymbol{\psi}) \| p(\phi^\tau)]. \quad (5)$$

We assume $q(\phi^\tau | \mathcal{D}^\tau; \boldsymbol{\psi})$ fully factorizes for each variable and also for each dimension as well:

$$q(\phi^\tau | \mathcal{D}^\tau; \boldsymbol{\psi}) = \prod_c q(\tilde{\omega}_c^\tau | \mathcal{D}^\tau; \boldsymbol{\psi}) \prod_l q(\tilde{\gamma}_l^\tau | \mathcal{D}^\tau; \boldsymbol{\psi}) \prod_i q(\tilde{z}_i^\tau | \mathcal{D}^\tau; \boldsymbol{\psi}) \qquad (6)$$

where we assume that each single dimension of $q(\phi^\tau | \mathcal{D}^\tau; \boldsymbol{\psi})$ follows univariate Gaussian having trainable mean and variance. We also let each dimension of prior $p(\phi^\tau)$ factorize into $\mathcal{N}(0, 1)$, such that the KL-divergence can have the especially simple closed form (Kingma & Welling, 2014).

The final form of the meta-training minimization objective with Monte-Carlo (MC) approximation for the expectation in Eq. (5) is as follows:

$$\min_{\boldsymbol{\theta}, \boldsymbol{\psi}} \frac{1}{M_\tau} \sum_{m=1}^{M_\tau} \frac{1}{S} \sum_{s=1}^{S} - \log p(\tilde{\mathbf{y}}_m^\tau | \tilde{\mathbf{x}}_m^\tau, \phi_s^\tau; \boldsymbol{\theta}) + \frac{1}{N_\tau + M_\tau} \mathrm{KL}[q(\phi^\tau | \mathcal{D}^\tau; \boldsymbol{\psi}) \| p(\phi^\tau)]. \quad (7)$$

where $\phi_s^\tau \sim q(\phi^\tau | \mathcal{D}^\tau; \boldsymbol{\psi})$. We implicitly assume the reparameterization trick for $\phi^\tau$ to obtain stable and unbiased gradient estimate w.r.t. $\boldsymbol{\psi}$ (Kingma & Welling, 2014). We set the MC sample size to $S = 1$ for meta-training for computational efficiency. When meta-testing, we perform MC approximation with sufficiently large sample size (e.g. $S = 10$):

$$p(\tilde{\mathbf{y}}_*^\tau | \tilde{\mathbf{x}}_*^\tau; \boldsymbol{\theta}) = \mathbb{E}_q[p(\tilde{\mathbf{y}}_*^\tau | \tilde{\mathbf{x}}_*^\tau, \phi^\tau; \boldsymbol{\theta})] \approx \frac{1}{S} \sum_{s=1}^{S} p(\tilde{\mathbf{y}}_*^\tau | \tilde{\mathbf{x}}_*^\tau, \phi_s^\tau; \boldsymbol{\theta}), \quad \phi_s^\tau \sim q(\phi^\tau | \mathcal{D}^\tau; \boldsymbol{\psi}). \quad (8)$$

or we may naively approximate by taking the expectation inside for computational efficiency:

$$\mathbb{E}_q[p(\tilde{\mathbf{y}}_*^\tau | \tilde{\mathbf{x}}_*^\tau, \phi^\tau; \boldsymbol{\theta})] \approx p(\tilde{\mathbf{y}}_*^\tau | \tilde{\mathbf{x}}_*^\tau, \mathbb{E}_q[\phi^\tau]; \boldsymbol{\theta}). \qquad (9)$$

In the experimental section, we show that Eq. (8) largely outperforms Eq. (9).

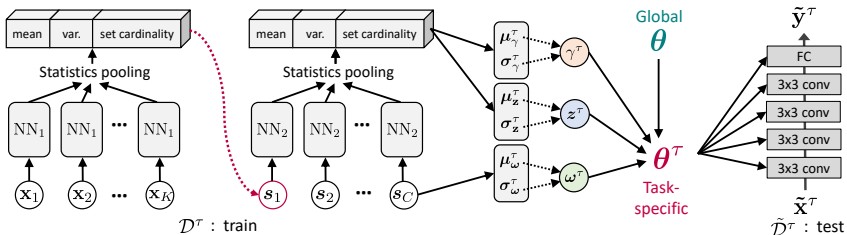

Figure 3: **Inference Network.** The proposed dataset encoder captures the instance-wise and class-wise statistics hierarchically, from which we infer three different balancing variables.

## 4.1 DATASET ENCODING

The main challenge in modeling our variational distribution $q(\phi^\tau | \mathcal{D}^\tau ; \psi)$ is to decide how to refine the training dataset $\mathcal{D}^\tau$ into informative representation, which is not trivial. This inference network should capture all the necessary statistical information to recognize any imbalances in the dataset $\mathcal{D}^\tau$. Mean-pooling (Edwards & Storkey, 2017) or sum-pooling (Zaheer et al., 2017) is frequently used as a practical set-encoder, where each instance in the set is transformed by the shared nonlinearity, and then averaged or summed together to generate a single vector summarizing the set, followed by an additional nonlinearity. However, for the classification dataset $\mathcal{D}^\tau$ which is the *set of (class) sets*, those non-hierarchical pooling methods will perform poorly as they ignore the label information. Therefore, we use a two-layer hierarchical set encoder which first encodes each class as a set of samples and then encodes the set as the set of classes (see the Appendix B for the justification).

However, there is an additional limitation of mean pooling when using it to describe a task: it does not recognize the number of elements in the set[2]. This could be a critical limitation in recognizing the imbalances in the given task. Therefore we explicitly input the number of elements into the set encoder. Yet, the set cardinality alone is insufficient in capturing the distribution of the dataset. Suppose that we have a set containing replications of a single instance. Then, although the set has only limited information, the set encoding cannot recognize it and will overestimate the information. To prevent this, we encode the variance of the set as well.

Based on this intuition, we define the set encoder $\mathrm{StatisticsPooling}(\cdot)$ that generates the concatenation of the set statistics such as mean, variance, and cardinality. We use this encoder to first encode each class, and then the whole set of classes as follows:

$$\mathbf{v}^\tau = \mathrm{StatisticsPooling}\left(\{\mathrm{NN}_2\left(\mathbf{s}_c\right)\}_{c=1}^C\right), \quad \mathbf{s}_c = \mathrm{StatisticsPooling}\left(\{\mathrm{NN}_1(\mathbf{x})\}_{\mathbf{x}\in\mathbf{X}_c^\tau}\right)$$

for classes $c = 1, \ldots, C$. $\mathbf{X}_c^\tau$ is the collection of class $c$ examples in task $\tau$. $\mathrm{NN}_1$ and $\mathrm{NN}_2$ are some neural networks parameterized by $\psi$. The vector $\mathbf{v}^\tau$ finally summarizes the entire dataset $\mathcal{D}^\tau$ for few-shot classification. Note that the class-specific balancing variables $\tilde{\omega}_1^\tau, \ldots, \tilde{\omega}_C^\tau$ are generated from $\mathbf{s}_1, \ldots, \mathbf{s}_C$, and other task-specific balancing variables $\tilde{\gamma}^\tau$ and $\tilde{\mathbf{z}}^\tau$ are generated from $\mathbf{v}^\tau$ with a few additional layers (See Figure 3).

## 5 EXPERIMENTS

We next validate our method on multiple benchmark datasets with more realistic task distribution.

**Datasets** We validate our method on the following benchmark datasets. **CIFAR-FS:** This dataset (Bertinetto et al., 2019) is a variant of CIFAR-100 dataset that consists of 100 general object categories. Each class comes with 600 examples, each of which is a color image that contains $32 \times 32$ pixels. We split the dataset into $64/16/20$ classes for training/validation/test. **SVHN:** This dataset (Netzer et al., 2011) is frequently used as an OOD dataset for CIFAR-10 and CIFAR-100. It consists of $26,032$ color images of $32 \times 32$ pixels, from 10 digits classes. **miniImageNet:** This dataset (Vinyals et al., 2016) is a subset of the ImageNet dataset (Russakovsky et al., 2015). It consists of total 100 classes, each of which has 600 images resized into $84 \times 84$. We split the dataset into subsets containing $64/16/20$ classes for training/validation/test. **CUB:** This dataset contains $11,788$ images of 200 bird species. We resize the images into $84 \times 84$. We split the dataset into

---

[2]We also found out that sum-pooling is very unstable in encoding set of variable size.

Table 1: **Any-shot classification results.** For each model, we run 3 independent trials and jointly test them over total $9,000 = 3 \times 3,000$ episodes. We report mean accuracies and $95\%$ confidence intervals.

| Meta-training | CIFAR-FS | | miniImageNet | |
|---|---|---|---|---|
| Meta-test | CIFAR-FS | SVHN | miniImageNet | CUB |
| MAML (Finn et al., 2017) | $71.55_{\pm 0.23}$ | $45.17_{\pm 0.22}$ | $66.64_{\pm 0.22}$ | $65.77_{\pm 0.24}$ |
| Meta-SGD (Li et al., 2017) | $72.71_{\pm 0.21}$ | $46.45_{\pm 0.24}$ | $69.95_{\pm 0.20}$ | $65.94_{\pm 0.22}$ |
| MT-net (Lee & Choi, 2018) | $72.30_{\pm 0.21}$ | $49.17_{\pm 0.21}$ | $67.63_{\pm 0.23}$ | $66.09_{\pm 0.23}$ |
| ABML (Ravi & Beatson, 2019) | $67.24_{\pm 0.24}$ | $36.52_{\pm 0.17}$ | $56.91_{\pm 0.19}$ | $57.88_{\pm 0.20}$ |
| Prototypical Networks (Snell et al., 2017) | $73.24_{\pm 0.20}$ | $42.91_{\pm 0.18}$ | $69.11_{\pm 0.19}$ | $60.80_{\pm 0.19}$ |
| Proto-MAML (Triantafillou et al., 2020) | $71.80_{\pm 0.21}$ | $40.16_{\pm 0.17}$ | $68.96_{\pm 0.18}$ | $61.77_{\pm 0.19}$ |
| Bayesian TAML | $\mathbf{75.15_{\pm 0.20}}$ | $\mathbf{51.87_{\pm 0.23}}$ | $\mathbf{71.46_{\pm 0.19}}$ | $\mathbf{71.71_{\pm 0.21}}$ |

Table 2: **Multi-dataset any-shot classification results.**

| Meta-training | Aircraft, QuickDraw, and VGG-Flower | | | | |
|---|---|---|---|---|---|
| Meta-test | Aircraft | QuickDraw | VGG-Flower | Traffic Signs | Fashion-MNIST |
| MAML | $48.60_{\pm 0.17}$ | $69.02_{\pm 0.18}$ | $60.38_{\pm 0.16}$ | $51.96_{\pm 0.22}$ | $63.10_{\pm 0.15}$ |
| Meta-SGD | $49.71_{\pm 0.17}$ | $70.26_{\pm 0.16}$ | $59.41_{\pm 0.27}$ | $52.07_{\pm 0.35}$ | $62.71_{\pm 0.25}$ |
| MT-net | $51.68_{\pm 0.17}$ | $68.78_{\pm 0.18}$ | $64.20_{\pm 0.16}$ | $56.36_{\pm 0.23}$ | $62.86_{\pm 0.15}$ |
| Prototypical Networks | $50.63_{\pm 0.16}$ | $\mathbf{72.31_{\pm 0.17}}$ | $65.52_{\pm 0.15}$ | $49.93_{\pm 0.18}$ | $64.26_{\pm 0.13}$ |
| Proto-MAML | $51.15_{\pm 0.17}$ | $69.84_{\pm 0.18}$ | $65.24_{\pm 0.17}$ | $53.93_{\pm 0.20}$ | $63.72_{\pm 0.15}$ |
| Bayesian TAML | $\mathbf{54.43_{\pm 0.16}}$ | $72.03_{\pm 0.16}$ | $\mathbf{67.72_{\pm 0.16}}$ | $\mathbf{64.81_{\pm 0.21}}$ | $\mathbf{68.94_{\pm 0.13}}$ |

100/50/50 classes for training/validation/test. Since this dataset is fine-grained, we regard it as an out-of-distribution dataset of the coarse-grained miniImageNet dataset.

**Realistic task distribution** To define realistic task distribution $p(\tau)$, we first randomly sample $C = 5$ classes from the whole set of classes. Then with $0.5$ probability, we sample the set size $N_c \sim \text{Unif}(1, 50)$ independently for each class $c = 1, \ldots, C$, in order to simulate *class imbalance*. On the other hand, with the other $0.5$ probability, we again sample the set size $N_c \sim \text{Unif}(1, 50)$ but apply the same single sample to all the classes, in order to simulate *task imbalance*. We set the number of test (or query) examples to 15 for each class.

**Experimental setup** We use conventional 4-block convolutional neural networks with 32 channels (Finn et al., 2017). All the baselines and our model become transductive through batch normalization at meta-test time, following Finn et al. (2017). We perform early-stopping for all the baselines and our model with meta-validation set performance. We set the number of inner-gradient steps to 5 for meta-training and 10 for meta-testing, for all the models that take inner-gradient steps. See the Appendix A for more details about the experimental setup.[3]

### 5.1 MAIN RESULTS

**Any-shot classification** Table 1 shows the results under the realistic task distribution with task and class imbalance. We first observe that Meta-SGD and MT-net outperforms MAML. They precondition the inner-gradients with diagonal or block-diagonal matrix (Flennerhag et al., 2020), which seems to provide extra flexibility to handle task and class imbalance. ABML, one of the recent Bayesian meta-learning models, largely underperforms MAML in this setting, mainly because each task-specific predictor is excessively regularized by the prior distribution. However, our graphical model in Figure 2 does not directly impose a prior distribution on task-specific predictors, but only on the balancing variables $\phi$. Prototypical Networks perform relatively well on in-distribution (ID) tasks but not on out-of-distribution (OOD) tasks. This is because metric-based models do not take the gradient steps for OOD tasks, such that the novel information in the dataset cannot be effectively incorporated into each task-specific predictor. Proto-MAML has been proposed to take the advantage of both metric-based and gradient-based approach (Triantafillou et al., 2020), but it does not outperform Prototypical Networks in our experiments. Finally, we observe that our Bayesian TAML significantly outperforms all the baselines on both ID and OOD datasets. Bayesian TAML performs especially well on OOD datasets, because when the given task largely differs from the meta-knowledge, the model is able to deviate far from the meta-knowledge; however this is difficult for the other baselines.

**Multi-dataset any-shot classification** We further test our model under a more challenging setting where tasks could come from a highly heterogeneous dataset (Triantafillou et al., 2020). We meta-

---

[3]Code is available at `https://github.com/haebeom-lee/l2b`

train the models with Aircrafts, Quickdraw and VGG-Flower dataset, and meta-test with the three datasets plus two additional datasets for OOD tasks - Traffic Signs and Fashion-MNIST. We see from Table 2 that Bayesian TAML also largely outperforms baselines in this setting. rototypical Networks perform relatively well for this multi-dataset classification due to their ability of learning a flexible metric space. Proto-MAML brings only marginal improvements on Prototypical Networks since it takes additional gradient steps without considering task dependency. On the other hand, Bayesian TAML effectively combines the advantage of both metric-based and gradient-based approaches by task-dependently modulating the gradient steps to handle task and class imbalance as well. See the Appendix A for the detailed experimental setup.

## 5.2 EFFECTIVENESS OF THE BALANCING VARIABLES

We now validate the effectiveness of the three balancing variables to clearly show their effectiveness.

$\mathbf{z}^\tau$ **for handling distributional shift.** $\mathbf{z}^\tau$ modulates the initial model parameter $\theta$, and decides what and how much to use from the meta-knowledge $\theta$ based on the relatedness between the initial $\theta$ and the task at hand. We validate the effectiveness of this balancing variable by examining the performance of the baseline methods and our models using the datasets (SVHN, CUB) that are highly heterogeneous from the meta-training datasets (CIFAR-FS, miniImageNet). The results in Table 3 show that our model with only $\mathbf{z}^\tau$ can effectively handle these out-of-distribution (OOD) tasks. We observe in Figure 4 that $\mathbf{z}^\tau$ actually relocates the initial $\theta$ far from the initial parameter for these OOD tasks given at meta-test time, with larger displacements for highly heterogeneous tasks (Figure 4, right). This allows the model to either stick to, or deviate from the meta-knowledge based on the similarity between the tasks given at the meta-training and meta-test time. Moreover, the last two rows of Table 3 show that the MC approximation in Eq. (8) largely outperforms the naive approximation in Eq. (9), which suggests that $\mathbf{z}^\tau$ learns very large variance. We conjecture that the role of such random initialization in MAML framework is to increase the effective learning rate for OOD tasks (See Appendix D for the discussion).

| Meta-training | CIFAR-FS | miniImageNet |
|---|---|---|
| Meta-test | SVHN | CUB |
| MAML | $45.17_{\pm 0.22}$ | $65.77_{\pm 0.24}$ |
| Meta-SGD | $46.45_{\pm 0.24}$ | $65.94_{\pm 0.22}$ |
| B. $\mathbf{z}$-TAML (naive approx.) | $47.80_{\pm 0.20}$ | $67.90_{\pm 0.21}$ |
| B. $\mathbf{z}$-TAML (MC approx.) | $\mathbf{52.29}_{\pm 0.24}$ | $\mathbf{69.11}_{\pm 0.22}$ |

Table 3: Ablation study on distributional shift.

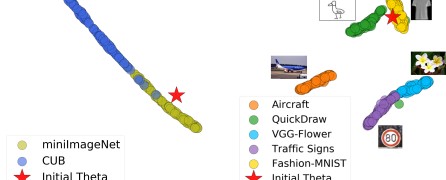

Figure 4: T-SNE visualization of $\theta$ and $\theta * \mathbb{E}[\mathbf{z}^\tau]$.

$\gamma^\tau$ **and** $\mathbf{z}^\tau$ **for handling task imbalance.** We then examine how the two balancing variables, $\gamma^\tau$ and $\mathbf{z}^\tau$ handle inter-task imbalance where tasks used for meta-learning have largely different number of instances. Figure 5(a) shows the performance of the baseline models and our models with each of the balancing variables, when the number of instances per task varies from 5 to 2000. We observe that our Bayesian $\mathbf{z}$-TAML and Bayesian $\gamma$-TAML largely outperform Meta-SGD, especially by large degree when the number of instances per task is large. Figure 5(b) shows that the effectiveness of $\mathbf{z}^\tau$ largely depends on the number of MC samples used in Eq. (8), demonstrating the importance of incorporating uncertainties in random initializations for handling task imbalance. We further observe from Figure 5(c) that the task-dependent learning rate multiplier $\gamma^\tau$ rapidly grows as the number of instances per task increases. This agrees with our intuition that larger tasks should take larger inner-gradient steps, to learn more from the given task rather than resorting to meta-knowledge.

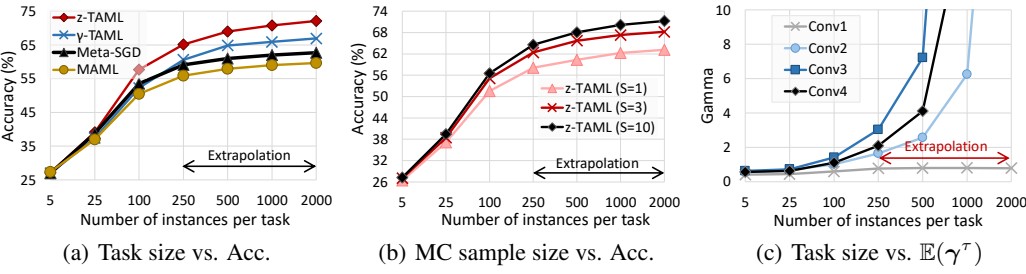

(a) Task size vs. Acc.  (b) MC sample size vs. Acc.  (c) Task size vs. $\mathbb{E}(\gamma^\tau)$

Figure 5: Ablation study on task imbalance (Meta-training: CIFAR-FS, Meta-test: SVHN).

$\omega^\tau$ **for handling class imbalance.** $\omega^\tau$ rescales the class-specific gradients to handle class imbalance where the number of instances per class (i.e. shot) largely varies. Table 4 shows the results under the varying degree of class imbalance across the task distribution. We observe that Bayesian $\omega$-TAML outperforms Meta-SGD, especially by larger degree under higher class imbalance. Notably, Bayesian $\omega$-TAML outperforms a heuristic balancing method which divides each class-specific gradient by the cardinality of each class set (Meta-SGD with $1/N$). The accuracy improvements over Meta-SGD in Figure 6 demonstrate that this heuristic balancing method overly emphasizes the tail classes with few training instances, thereby deteriorating the performance on classes with sufficiently large number of instances. On the other hand, our Bayesian $\omega$-TAML learns the appropriate balancing variables which allow to obtain large performance gains on all classes.

| Meta-training / Meta-test CIFAR-FS / CIFAR-FS | Number of instances per class | | |
|---|---|---|---|
| | 10 | 5 - 25 | 1 - 50 |
| MAML | **73.60**±0.19 | 71.15±0.19 | 67.43±0.22 |
| Meta-SGD | 73.25±0.19 | 72.68±0.19 | 71.61±0.19 |
| Meta-SGD with $1/N$ | 71.33±0.19 | 72.43±0.19 | 72.23±0.19 |
| Bayesian $\omega$-TAML | 73.44±0.18 | **73.20**±**0.18** | **72.86**±**0.19** |

Table 4: Ablation study on class imbalance.

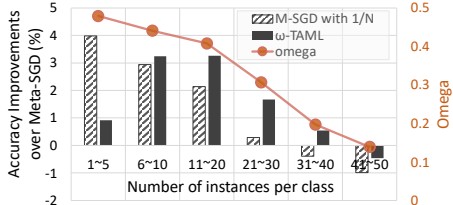

Figure 6: $\mathbb{E}[\omega^\tau]$ and accuracy improvements over Meta-SGD.

### 5.3 MORE ABLATION STUDIES

**Effectiveness of Bayesian modeling** We further demonstrate the effectiveness of Bayesian modeling by comparing it with the deterministic version of our model (Deterministic TAML), where three balancing variables are no longer stochastic. We apply $\ell_2$ regularization on the variables with coefficients that are equivalent to the KL-divergence in Eq. (5). The results in Table 5 show that the Bayesian TAML significantly outperforms its deterministic counterpart, especially with larger gap on the OOD task (SVHN). We also see from the last two rows of the table that MC approximation in Eq. (8) is more beneficial for the OOD tasks than for the ID tasks (See Appendix D for the discussion). Figure 7 further shows that the balancing variable $\gamma^\tau$ for handling task imbalance, more sensitively reacts on Bayesian TAML than on Deterministic TAML, which suggests that Bayesian modeling amplifies the effect of the balancing variables.

| Meta-training: CIFAR-FS | CIFAR-FS | SVHN |
|---|---|---|
| MAML | 70.19±0.23 | 41.81±0.19 |
| Meta-SGD | 72.71±0.21 | 46.45±0.24 |
| Deterministic TAML | 73.82±0.21 | 46.78±0.24 |
| Bayesian TAML (Naive approx.) | 73.52±0.20 | 47.15±0.20 |
| Bayesian TAML (MC approx.) | **75.15**±**0.20** | **51.87**±**0.23** |

Table 5: Classification performance of Bayesian and Deterministic TAML on seen and unseen dataset.

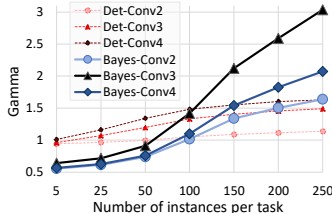

Figure 7: $\mathbb{E}[\gamma^\tau]$ vs. Bayesianness.

**Dataset encoding** Finally, we perform an ablation study of the proposed task encoding network. The results in Table 6 show that the proposed hierarchical encoding scheme for classification dataset, with set cardinality and variance is significantly more effective than simple mean-pooling methods (Zaheer et al., 2017; Edwards & Storkey, 2017)[4].

| Meta-training / Meta-test CIFAR-FS / CIFAR-FS | Hierarchical encoding | |
|---|---|---|
| | × | √ |
| Mean | 73.84±0.21 | 73.69±0.21 |
| Mean + N | 73.17±0.21 | 74.88±0.20 |
| Mean + Var. + N | 73.93±0.21 | **75.15**±**0.20** |

Table 6: Ablation study on dataset encoding schemes. N: Set cardinality.

## 6 CONCLUSION

We propose Bayesian TAML that learns to balance the effect of meta-learning and task-adaptive learning, to consider meta-learning under a more realistic task distribution where each task and class can have varying number of instances. Specifically, we encode the dataset for each task into

---

[4]We also found out that using set cardinality is more effective than utilizing higher-order statistics such as skewness or kurtosis (See the Appendix C for further ablation study).

hierarchical representations, and use it to modulate the original parameter, learning rate, and the class-specific gradients. We use a Bayesian framework to infer the posterior of these balancing variables, and propose an effective variational inference framework to solve for them. Our model outperforms existing meta-learning methods when validated on imbalanced few-shot classification tasks. Further analysis of each balancing variable shows that each variable effectively handles task imbalance, class imbalance, and out-of-distribution tasks. We believe that our work makes a meaningful step toward application of meta-learning to real-world problems.

**Acknowledgements**    This work was supported by Google AI Focused Research Award, Samsung Research Funding Center of Samsung Electronics (No. SRFC-IT1502-51), the Engineering Research Center Program through the National Research Foundation of Korea (NRF) funded by the Korean Government MSIT (NRF-2018R1A5A1059921), and the Institute of Information & communications Technology Planning & Evaluation (IITP) grant funded by the Korea government(MSIT) (No.2019-0-00075, and Artificial Intelligence Graduate School Program (KAIST)).

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

# A  EXPERIMENTAL SETUP

## A.1  BASELINES

We describe the baseline models and our task-adaptive learning to balance model.

**1) MAML.** The Model-Agnostic Meta-Learning (MAML) model by Finn et al. (2017), which aims to learn the global initial model parameter, from which we can take a few gradient steps to get task-specific predictors.

**2) ABML.** This model interprets MAML under hierarchical Bayesian framework, but they propose to share and amortize the inference rules across both global initial parameters as well as the task-specific parameters.

**3) MT-net.** A gradient-based meta-learning model proposed by Lee & Choi (2018). The model obtains a task-specific parameter only w.r.t. a subset of the whole dimension (M-net), followed by a linear transformation to learn a metric space (T-net).

**4) Meta-SGD.** A base MAML with the learnable learning-rate vector (without any restriction on sign) element-wisely multiplied to each step inner-gradient (Li et al., 2017).

**5) Prototypical Networks.** A metric-based few-shot classification model proposed by Snell et al. (2017). The model learns a metric space based on Euclidean distance between class prototypes and query embeddings.

**6) Proto-MAML.** A variant of MAML (Triantafillou et al., 2020) that replaces the initialization of the final fully-connected layer matrix with the equivalent one of the Prototypical Networks (Snell et al., 2017). This model combines the advantage of both metric-based and gradient-based approach. We set $\alpha$ to $0.0005$ for any-shot classification and $0.01$ for multi-dataset experiments.

**7) Bayesian TAML.** Our model that can adaptively balance between meta- and task-specific learners for each task and class.

## A.2  ANY-SHOT CLASSIFICATION.

We describe more detailed experimental settings for any-shot classification. For MAML, ABML and MT-NET, we set the inner-gradient stepsize $\alpha$ to $0.5$ for CIFAR-FS / SVHN, and $0.1$ for miniImageNet / CUB, after searching the range $\alpha \in \{0.01, 0.05, 0.1, 0.5\}$.

**CIFAR-FS and SVHN:**  We meta-train all models for total $50K$ iterations with meta-batch size set to $4$. The outer learning rate is set to $0.001$ for all the baselines and our models.

**miniImageNet and CUB:**  We meta-train all models for total $80K$ iterations with meta-batch size set to $1$. The outer learning rate is set to $0.0001$ for all the baselines and our models.

## A.3  MULTI-DATASET CLASSIFICATION

For multi-dataset classification, we construct a subset of the whole collection of the Meta-Dataset (Triantafillou et al., 2020). We resize the images into $32 \times 32$ pixels. For each task, We uniformly select one dataset among Aircrafts, Quickdraw and VGG-Flower and randomly sample 10 classes from the dataset. Then we sample instances from each class with the number of instances per class ranging from 1 to 50. The number of test instances is equally set to 15 for all classes. At meta-test time, we use the three datasets plus two more out-of-distribution datasets - Traffic Signs and Fashion-MNIST. For MAML, ABML and MT-NET, we set the inner-gradient stepsize $\alpha$ to $0.5$. We set the number of classes for each task to 10, meta-batch size to 3, meta-training iterations to $60K$, and outer learning rate to $0.001$ for all models.

**Aircraft:**  We split this dataset (Maji et al., 2013) into $70/15/15$ classes for meta- training/validation/test with 100 examples for each class.

**Quick Draw:**  We split this dataset (Jonas Jongejan & Fox-Gieg, 2016) into $241/52/52$ classes for meta- training/validation/test with randomly sampled 200 examples for each class.

**VGG Flower:**  This dataset (Nilsback & Zisserman, 2008) contains 40 between 258 images for each class and we split this dataset into $71/16/15$ classes for meta- training/validation/test.

**Traffic Signs:** This dataset (Houben et al., 2013) has only test set consisting of 43 classes. Each class has 900 examples.

**Fashion-MNIST:** We use the test set of Fashion-MNIST (Xiao et al., 2017) for meta-testing. This dataset has 10 classes with 1000 examples per class.

### A.4 INFERENCE NETWORK ARCHITECTURE

We describe the network architecture of the inference network that takes a classification dataset as an input and generates three balancing variables as output.

**Shared encoder** $\mathrm{NN}_1 : \mathbf{X}_1^\tau, \ldots, \mathbf{X}_C^\tau \to \mathbf{s}_1, \ldots, \mathbf{x}_C$

*3 × 3 Conv2d* with 10-dim and ReLU
*2 × 2 Max pool*
*3 × 3 Conv2d* with 10-dim and ReLU
*2 × 2 Max pool*
*fc* layer with 64-dim
*Statistics Pooling* for each class $c = 1, \ldots, C$.
*fc* layer with 4-dim (across the statistics) and ReLU

**Shared encoder** $\mathrm{NN}_2 : \mathbf{s}_1, \ldots, \mathbf{s}_C \to \mathbf{v}^\tau$

*fc* layer with 128-dim and ReLU
*fc* layer with 32-dim
*Statistics Pooling* over all the class representations
*fc* layer with 4-dim (across the statistics) and ReLU

**Generate** $\boldsymbol{\omega}^\tau : \mathbf{s}_1^\tau, \ldots, \mathbf{s}_C^\tau \to (\mu_{\omega_1}^\tau, \sigma_{\omega_1}^\tau), \ldots, (\mu_{\omega_C}^\tau, \sigma_{\omega_C}^\tau)$

*fc* layer with 64-dim and ReLU
*fc* layer to generate $(\mu_{\omega_c}^\tau, \sigma_{\omega_c}^\tau)$ for each class $c = 1, \ldots, C$.

**Generate** $\boldsymbol{\gamma}^\tau : \mathbf{v}^\tau \to (\mu_{\gamma_1}^\tau, \sigma_{\gamma_1}^\tau), \ldots, (\mu_{\gamma_L}^\tau, \sigma_{\gamma_L}^\tau)$

*fc* layer with 64-dim and ReLU
*fc* layer to generate $(\mu_{\gamma_l}^\tau, \sigma_{\gamma_l}^\tau)$ for each layer $l = 1, \ldots, L$.

**Generate** $\mathbf{z}^\tau : \mathbf{v}^\tau \to \boldsymbol{\mu}_\mathbf{z}^\tau, \boldsymbol{\sigma}_\mathbf{z}^\tau$

*fc* layer with 64-dim and ReLU
*fc* layer to generate $(\boldsymbol{\mu}_\mathbf{z}^\tau, \boldsymbol{\sigma}_\mathbf{z}^\tau)$ for the output channels

## B    JUSTIFICATION FOR SET-OF-SETS STRUCTURE.

Based on the previous justification of DeepSets (Zaheer et al., 2017), we can easily justify the Set-of-Sets structure proposed in the main paper as well, in terms of the two-level permutation invariance properties required for any classification dataset. The main theorem of DeepSets is:

**Theorem 1.** *A function $f$ operating on a set $\mathbf{X} \in \mathcal{X}$ is a valid set function (i.e. permutation invariant), iff it can be decomposed as $f(\mathbf{X}) = \rho_2(\sum_{\mathbf{x} \in \mathbf{X}} \rho_1(\mathbf{x}))$, where $\rho_1$ and $\rho_2$ are appropriate nonfcities.*

See (Zaheer et al., 2017) for the proof. Here we apply the same argument twice as follows.

1. A function $f$ operating on a set of representations $\{\mathbf{s}_1, \ldots, \mathbf{s}_C\}$ (we assume each $\mathbf{s}_c$ is an output from a shared function $g$) is a valid set function (i.e. permutation invariant w.r.t. the order of $\{\mathbf{s}_1, \ldots, \mathbf{s}_C\}$), *iff* it can be decomposed as $f(\{\mathbf{s}_1, \ldots, \mathbf{s}_C\}) = \rho_2(\sum_{c=1}^C \rho_1(\mathbf{s}_c))$ with appropriate nonfcities $\rho_1$ and $\rho_2$.

2. A function $g$ operating on a set of examples $\{\mathbf{x}_{c,1}, \ldots, \mathbf{x}_{c,N}\}$ is a valid set function (i.e. permutation invariant w.r.t. the order of $\{\mathbf{x}_{c,1}, \ldots, \mathbf{x}_{c,N}\}$) *iff* it can be decomposed as $g(\{\mathbf{x}_{c,1}, \ldots, \mathbf{x}_{c,N}\}) = \rho_4(\sum_{i=1}^{N} \rho_3(\mathbf{x}_{c,i}))$ with appropriate nonfcities $\rho_3$ and $\rho_4$.

Inserting $\mathbf{s}_c = g(\{\mathbf{x}_{c,1}, \ldots, \mathbf{x}_{c,N}\})$ into the expression of $f$, we arrive at the following valid composite function operating on a set of sets:

$$f\left(\{g(\{\mathbf{x}_{c,1}, \ldots, \mathbf{x}_{c,N}\})\}_{c=1}^{C}\right) = \rho_2\left(\sum_{c=1}^{C} \rho_1\left(\rho_4\left(\sum_{i=1}^{N} \rho_3(\mathbf{x}_{c,i})\right)\right)\right) \tag{10}$$

Let $F$ denote the composite of $f$ and (multiple) $g$ and let $\mathrm{NN}_2$ denote the composite of $\rho_1$ and $\rho_4$. Further define $\mathrm{NN}_1 := \rho_3$ and $\mathrm{NN}_3 := \rho_2$. Then, we have

$$F\left(\{\{\mathbf{x}_{1,1}, \ldots, \mathbf{x}_{1,N}\}, \ldots, \{\mathbf{x}_{C,1}, \ldots, \mathbf{x}_{C,N}\}\}\right) = \mathrm{NN}_3\left(\sum_{c=1}^{C} \mathrm{NN}_2\left(\sum_{i=1}^{N} \mathrm{NN}_1(\mathbf{x}_{c,i})\right)\right) \tag{11}$$

where $C$ is the number of classes and $N$ is the number of examples per class. See Section A.4 for the actual encoder structure.

## C   ABLATION STUDY ON HIGHER-ORDER STATISTICS

While our framework does not place any restriction on selecting the statistics for set encoding, we perform further ablation study on the effectiveness of higher-order statistics for our specific experimental setting. We see from Table 7 that the higher-order statistics such as element-wise sample skewness and kurtosis improve the performance given sample mean and diagonal covariance. However, the set cardinality seems more effective than those statistics as it could be the most direct and relevant criteria for detecting imbalances in a set.

|  | Meta-training | Meta-test | |
|---|---|---|---|
|  | CIFAR-FS | CIFAR-FS | SVHN |
| Mean + Var. | | $73.37_{\pm 0.21}$ | $49.81_{\pm 0.22}$ |
| Mean + Var. + Skew. | | $73.66_{\pm 0.21}$ | $50.33_{\pm 0.23}$ |
| Mean + Var. + Kurt. | | $73.47_{\pm 0.21}$ | $50.27_{\pm 0.23}$ |
| Mean + Var. + N | | $\mathbf{75.15_{\pm 0.20}}$ | $\mathbf{51.87_{\pm 0.23}}$ |

Table 7: Further ablation study on dataset encoding schemes. N: Set cardinality.

## D   ANALYSIS ON RANDOM INITIALIZATION FOR SOLVING OUT-OF-DISTRIBUTION TASKS

In this section, we analyze the effect of the variance of $\mathbf{z}^\tau$ for solving OOD tasks, that randomizes the MAML initialization parameter $\boldsymbol{\theta}$. Define $\mathbb{E}_{q(\mathbf{z}^\tau | \mathcal{D}^\tau; \boldsymbol{\psi})}[\boldsymbol{\theta}_0] = \boldsymbol{\theta} * \mathbb{E}_q[\mathbf{z}^\tau]$, the initialization parameter modulated by the posterior mean of $\mathbf{z}^\tau$. Then, we measure the $\ell_2$ distance from $\mathbb{E}_q[\boldsymbol{\theta}_0]$ to the two different versions of the final task-specific parameter $\boldsymbol{\theta}^\tau$ after taking 10 gradient steps, in order to see how the posterior variance of $\mathbf{z}^\tau$ affects the final task-specific parameter $\boldsymbol{\theta}^\tau$ as a function of $\mathbf{z}^\tau$:

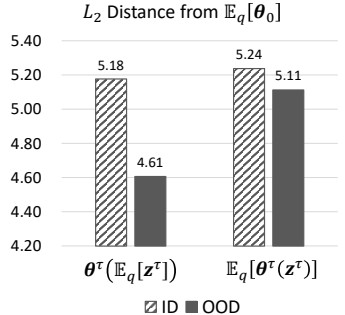

Figure 8: $\ell_2$ distance between the initialization and the task-specific parameters, under different treatment of the expectation over $q(\mathbf{z}^\tau | \mathcal{D}^\tau; \boldsymbol{\psi})$. We use Bayesian $\mathbf{z}$-TAML and evaluate with CIFAR-FS / SVHN 50-shot tasks.

- $\boldsymbol{\theta}^\tau(\mathbb{E}_q[\mathbf{z}^\tau])$: Task-specific predictor $\boldsymbol{\theta}^\tau$ obtained *without* the variance of $\mathbf{z}^\tau$, such that the expectation is taken *before* the inner-gradient steps.

- $\mathbb{E}_z[\boldsymbol{\theta}^\tau(\mathbf{z}^\tau)]$: Task-specific predictor $\boldsymbol{\theta}^\tau$ obtained *with* the variance in the random initialization, such that the expectation is taken *outside of* the gradient steps. We evaluate the expectation with MC approximation, having the ensemble of the diverse task-specific predictor samples $\boldsymbol{\theta}_1^\tau, \ldots, \boldsymbol{\theta}_S^\tau$ (we use $S = 50$).

Figure 8 suggests that the role of the random initialization is to *increase the effective learning rate for the OOD tasks*. We see from the left bar graph that if we do not consider variance in the initialization ($\boldsymbol{\theta}^\tau(\mathbb{E}_q[\mathbf{z}^\tau])$), the OOD tasks deviate relatively less than the ID tasks (4.61 vs. 5.18), although it

should deviate much considering the distributional discrepancy. On the other hand, if we incorporate the random initialization to obtain task-specific parameter ($\mathbb{E}_q[\boldsymbol{\theta}^\tau(\mathbf{z}^\tau)]$), OOD tasks can deviate further from the initialization ($4.61 \rightarrow 5.11$). It directly results in the performance gain because the task-specific learner can exploit more of the information in the OOD tasks.

