# OpenReview forum: "Learning to Balance: Bayesian Meta-Learning for Imbalanced and Out-of-distribution Tasks"
_ICLR.cc/2020/Conference — Accept (Talk)_

### Official Review · AnonReviewer2 · 2019-10-21
**Official Blind Review #2**

**Rating:** 8

**Review:**

Summary
-------------
This paper proposed to improve existing meta learning algorithms in the presence of task imbalance, class imbalance, and out-of-distribution tasks. Starting from the model-agnostic meta-learning (MAML) algorithm (Finn et al. 2017), to tackle task imbalance, where the number of training examples of varies across different tasks, a task-dependent learning rate decaying factor was learned to be large for large tasks and small for small tasks. In this way, the small task can benefit more from the meta-knowledge and the large task can benefit more from task-specific training. To tackle class imbalance, a class-specific scaling factor was applied to the class-specific gradient. The scaling factor was large for small class and small for large class so that different classes can be treated equally. To tackle the out-of-distribution tasks, a task-dependent variables was learned to emphasize meta-knowledge for the test task similar to training tasks. Additional model parameters are learned through variational inference. Experimental results on benchmark datasets demonstrate the proposed approach outperformed its competing alternatives. Analysis of each component confirm they work as expected.

Comments
---------------
This paper is well motivated and clearly written. The empirical evaluation also support major claims in the paper.

Can the author provide more details on the inference of the model? In the likelihood term in Eq. (7), the task specific parameters \theta^{\tau} was parameterized by Eq. (3), which contains K iterative gradient updates. How was the gradient w.r.t. \theta was computed in this setting?

The task-specific learning rate decaying factor was constrained to be between 0 and 1 using the function f(). The class-specific scaling factor made use of the SoftPlus() function, for the same purpose of scaling learning rate, why do these two different options of functions were applied?

For the scaling vector of the initial parameters g(z^{\tau}), for its zero entries, the initialization of the corresponding entries in task-specific parameter \theta would be zero. Would it be better to apply a linear interpolation between \theta and a randomly-initialized vector in Eq (2)?

Edits after reading the author's rebuttal
==================================
The author's reply well addressed my questions. After reading other reviewers' positive comments and the author's thorough reply, I decide to increase my rating to 8: Accept.

**Experience Assessment:**

I do not know much about this area.

**Review Assessment: Checking Correctness Of Derivations And Theory:**

I carefully checked the derivations and theory.

**Review Assessment: Checking Correctness Of Experiments:**

I carefully checked the experiments.

**Review Assessment: Thoroughness In Paper Reading:**

I read the paper thoroughly.

---

> ### Author Response · Authors · 2019-11-15
> **Response to Reviewer #2**
>
> We really appreciate your constructive comments. We respond to each comment as follows.
>
> 1. How was the gradient w.r.t. $\theta$ was computed?
>
> - Basically, computation of gradients w.r.t. initial $\theta$ in our model is exactly the same as the original MAML framework. Suppose that the number of inner-gradient steps is 5 and k-th inner-gradient update is denoted as a function $u_k$. Then, from the initial model parameter $\theta_0$, we have
>
> $\theta_1 = u_1(\theta_0)$
> $\theta_2 = u_2(\theta_1)$
> …
> $\theta_5 = u_5(\theta_4)$
> $\text{Loss_test} = \mathcal{L}(\theta_5, \mathcal{D}_\text{test})$
>
> Thus it is trivial to compute the gradient of Loss_test w.r.t. $\theta_0$ using the chain rule.
>
>
> 2. Why constrain the task-specific learning rate decaying factor to a value between 0 and 1, while using softplus() that does not have such a constraint for the class-specific scaling factor?
>
> - Note that the task-specific learning rate decays as follows:
>
> $\alpha$, $\alpha f(\gamma)$, $\alpha f(\gamma)^2$, $\alpha f(\gamma)^3$, …
>
> Without such restriction to force it between 0 and 1, the learning rate may largely diverge due to its exponential form.
>
> On the other hand, since the class-specific scaling factor is simply a multiplicative coefficient, there is no such restriction, except for the positivity constraint. Thus we used softplus function to model the class-specific scaling factor.
>
>
> 3. Applying a linear interpolation between $\theta$ and a randomly-initialized vector in Eq (2) considering zero-entries caused by $g(z^{\tau})$.
>
> - Thanks you for your insightful suggestion. As you mentioned, applying $g(z)$ to the shared initial parameter $\theta$ may set some of the initial parameters to smaller values. However, this is not problematic, but rather beneficial since the parameters in the shared $\theta$ that are irrelevant for the given task need to be properly suppressed such that it does not distract the learning on the current task.
>
> However, the interpolation technique you mentioned, which learns to interpolate between a random weight matrix and $\theta$ ($\theta_0 = \theta\odot g(z) + (1-g(z)) \odot \theta_{rand}$, where $\odot$ is element-wise multiplication and $\theta_{rand}$ is a random weight matrix) may have a similar effect, and we performed experimental validation of it during the rebuttal period.
>
> Models					                    Omniglot	        MNIST
> Meta-SGD				                    98.38+-0.23	90.42+-0.28
> Bayesian g(z) TAML + Interpolation	    98.83+-0.08	92.03+-0.19
> Bayesian g(z) TAML			            98.92+-0.30	92.06+-0.32
>
> The results show that Bayesian TAML with the interpolation technique (Bayesian g(z) TAML + Interpolation) have a very similar performance as the original Bayesian TAML with the masking scheme. Thus the two techniques seem to have a similar effect, on deciding what to use from the meta-knowledge and what to forget.
>
> We included these results in Appendix F.

---

### Official Review · AnonReviewer3 · 2019-10-22
**Official Blind Review #3**

**Rating:** 8

**Review:**

The paper proposes a Bayesian approach for meta learning in settings were the tasks might be OOD or have imbalanced class distribution. The proposed approach has 3 task-specific balancing variables with a prior and an inference network. Using an amortized inference scheme, the model unifies the meta-learning objective loss with the lower bound of probabilistic model marginal likelihood.

The paper is well-written and well-motivated. I only have some minor comments and questions:
- Can you add the standard errors for the results in Section 5.2 (maybe at least in the Appendix)?
- Specifically it would be interesting to see if the results for analyzing the class imbalance variable are statistically significant; specially in light of the recent work on the effects of importance weighting in DL (see “What is the Effect of Importance Weighting in Deep Learning?” by Byrd and Lipton) which essentially question the value of importance weighting for handling class imbalance in various DL settings.
- For the experiments in Section 5.1 can you also report the values for the three task-specific balancing variables? (Maybe in the Appendix).

Minor:
“obtains significantly improves over” -> "significantly improves over"

Overall, I found the paper interesting and practically useful, although I believe some additions to the empirical evaluation can improve the impact of the paper.

**Experience Assessment:**

I have read many papers in this area.

**Review Assessment: Checking Correctness Of Derivations And Theory:**

I assessed the sensibility of the derivations and theory.

**Review Assessment: Checking Correctness Of Experiments:**

I assessed the sensibility of the experiments.

**Review Assessment: Thoroughness In Paper Reading:**

I read the paper at least twice and used my best judgement in assessing the paper.

---

> ### Author Response · Authors · 2019-11-15
> **Response to Reviewer #3**
>
> We really appreciate your constructive comments. We respond to each comment as follows.
>
> 1.  Including standard errors for the results in Section 5.2.
>
> - We omitted them due to space limitation. As you suggested, we added in the standard errors to all the tables in the Experiment section, including the tables in Section 5.2 in the revision.
>
> 2. It would be interesting to see the results of analyzing the class imbalance variables are statistically significant, specially in light of a recent work by Byrd and Liption [1]
>
> - Based on the standard errors we added into the paper (See Table 5), the effectiveness of class-imbalance variables are indeed statistically significant.
>
> - Thank you for letting us know a very interesting paper [1]. It is interesting that for DNNs of high capacity, the effect of importance weighting asymptotically disappears if no regularizers are applied. However in Figure 6, we can see that the per-class improvements are higher as the number of instances becomes smaller, which suggests that in our case, the class balancing variable $g(\omega)$ is still effective in fixing class imbalance.
>
> - We conjecture that such correct behavior of $g(\omega)$ is an effect of early-stopping, which is enforced to any MAML-variants by design [2]. Byrd and Lipton [1] mention that early-stopping prevents the effect of importance sampling from disappearing (Figure 5 and Figure 6 in [1]). Since all MAML-variants, including our model, only take few gradient steps for each task, our model does not get affected by such vanishing effect. This early-stopping behavior is actually further promoted in our model, as we have an additional learning-rate decaying factor $f(\gamma)$ which reduces the total length of the learning trajectory.
>
> = Reference =
> [1] Byrd and Lipton., What is the Effect of Importance Weighting in Deep Learning?, ICML, 2019.
> [2] Grant et al., Recasting Gradient-Based Meta-Learning as Hierarchical Bayes, ICLR, 2018.
>
>
> 3. Can you report the values of the balancing variables of Section 5.1 to the Appendix?
>
> - We report the values of the balancing variables with the full model in Section E of the appendix. The trends on the three balancing variables are similar to what we report in the ablation study.

---

### Official Review · AnonReviewer1 · 2019-10-23
**Official Blind Review #1**

**Rating:** 8

**Review:**

Summary
========
This paper introduces a mechanism for gradient-based meta-learning models for few-shot classification to be able to adapt to diverse tasks that are imbalanced and heterogeneous. In particular, each encountered task may have varying numbers of shots (task imbalance) and even within each task, different classes may have different numbers of shots (class imbalance). Further, test tasks might come from a different distribution than the training tasks. They propose to handle this scenario by introducing three new types of variables which control different facets of the degree and type of task adaptation, allowing to decide how much to reuse meta-learned knowledge versus new knowledge acquired from the training set of the given task.

Specifically, their newly introduced variables are: 1) the factor for learning rate decay (a scalar) for the inner-loop task adaptation optimization which allows to not deviate too much from the global initialization when insufficient data is available, 2) a class-specific learning rate (one scalar per class) that allows to tune more for under-represented classes of the training set, 3) a set of weights on the global initialization (one scalar per dimension) that can down-weigh each component if it’s not useful for the task at hand (e.g. if test tasks have significantly different statistics than training tasks did).

The values of these variables are predicted based on the training set of the task: the support set is encoded via a hierarchical variant of a set encoding (where pooling is done using higher order statistics too instead of simply averaging). The resulting encoded support set is the input to the network that produces the values for the three sets of variables discussed above. Each new variable is treated in a Bayesian fashion: a prior is defined over it (Normal(0,1)), which is updated by conditioning on the training set to form a posterior for each given task. Specifically, each of the above variables is represented by a Gaussian whose mean and variance are the learnable parameters that are produced by the network described above.

Experimentally, this method outperforms others on a setting of imbalanced tasks (the shot is sampled uniformly at random from a designated range). The gain over other methods is large in particular when evaluated on out-of-distribution tasks (coming from a different dataset) and when the imbalance is large.

Comments (in decreasing order of importance)
========================================
A) The Bayesian framework helps because it offers an elegant way to use a prior. In the deterministic version, was any effort made to resemble the effect of that prior? For example, one can define a regularizer that penalizes behaviors that ignore the meta-knowledge too much (e.g. too large values for \gamma, or for the class-specific learning rates etc). Albeit more ‘hacky’, if these regularization coefficients are tuned properly, they might result in a similar effect to that of having a prior. A fair comparison to the deterministic variant should include this.

B) I think that \gamma and z can be merged into a single set of parameters? In particular, imagine a per-dimension-of-\theta learning rate. This would then be large for a dimension when there is a larger need for adapting that dimension of \theta. In the case of large training sets, this can be large for all dimensions, recovering the behavior of a large \gamma. For the case of diverse datasets, this would behave as the current z (updates a lot the dimensions of \theta that are irrelevant for the given task due to the dataset shift).

C) Meta-Dataset (https://arxiv.org/abs/1903.03096) is a recent benchmark for few-shot classification that introduces both of what is referred to here as task imbalance and class imbalance and also is comprised of heterogeneous datasets and evaluates performance on some held-out datasets too. The current state-of-the-art on it (as far as I know) is CNAPs [1] which employs a flexible adaptation mechanism on a per-task basis but is fully amortized (performs no gradient-based adaptation to each task) and makes no explicit effort to tackle imbalanced tasks as is done here. I’m curious how this method would compete in that setup. It definitely seems to be a strong candidate for that benchmark!

Less important
=============
D) which dataset is used in Tables 4 and 5? I assume it’s Omniglot (due to the numbers being in the 90s) but it would be good to say this explicitly.
E) In section 5.2, expressions such as x5 and x15 are used to characterize the degree of imbalance of a task. How exactly are these computed? Does x5 mean that the largest shot is 5 times larger than the smallest shot? It would be good to explicitly state this.
F) In the Related Work section, in the Meta-learning paragraph there is a sentence that’s not accurate: “Metric-based approaches learn a shared metric space [...] such that the instances are closer to their correct prototypes than to others”. This sentence does not describe all metric based approaches. It describes Prototypical Networks (Snell et al) but not, for example, Matching Networks (Vinyals et al) nor many others that like Matching Networks perform example-based comparisons and don’t aggregate a class’ examples into a prototype.

In a nutshell
===========
I think this work is a useful contribution for moving towards a more realistic setting in few-shot classification. It captures some desiderata of models that can operate in more realistic settings and outperforms previous models in those scenarios. My comments above are mostly suggesting improvements and clarifications but I am inclined to recommend acceptance of this paper.

References
=========
[1] Fast and Flexible Multi-Task Classification Using Conditional Neural Adaptive Processes. Requeima et al. NeurIPS 2019.


**Experience Assessment:**

I have published one or two papers in this area.

**Review Assessment: Checking Correctness Of Derivations And Theory:**

N/A

**Review Assessment: Checking Correctness Of Experiments:**

I assessed the sensibility of the experiments.

**Review Assessment: Thoroughness In Paper Reading:**

I read the paper thoroughly.

---

> ### Author Response · Authors · 2019-11-15
> **Response to Reviewer #1**
>
> We really appreciate your constructive comments. We respond to each comment as follows.
>
> 1. Comparison between Bayesian TAML and regularized Deterministic TAML.
>
> - Thank you for your helpful suggestion. As you suggested, we applied L2 regularization on the three balancing variables, with coefficients {1e-1,1e-2,1e-3,1e-4}. Note that since the inference network is shared for three balancing variables, the regularized version of deterministic TAML only introduce a single hyperparameter to tune. We named the model as Deterministic TAML (L2 Reg.). The results are as follows.
>
> Models					        Omniglot	MNIST
> MAML					        98.32+-0.26	89.19+-0.31
> Determ. TAML (No Reg.)		98.75+-0.34	90.77+-0.36
> Determ. TAML (L2 Reg., 1e-1)    98.98+-0.08    85.92+-0.31
> Determ. TAML (L2 Reg., 1e-2)	98.85+-0.08	90.97+-0.22
> Determ. TAML (L2 Reg., 1e-3)	98.93+-0.09	91.52+-0.21
> Determ. TAML (L2 Reg., 1e-4)	98.58+-0.10	91.13+-0.22
> Bayesian TAML			        99.13+-0.31	92.38+-0.33
>
> We observe that applying weight decay regularization to the balancing variables of Deterministic TAML improves the model performance. However, it may even underperform the original MAML if the hyperparameter $\lambda$ is not tune with cross-validation (e.g. $\lambda = 1e-1$). Bayesian TAML does not require any additional hyperparameter tuning, and still largely outperform this deterministic version of TAML. In the revision, we added in the performance of this regularized version of Deterministic TAML, in Table 6.
>
> 2. Effectiveness of merging $\gamma$ which handles task imbalance and $z$ which tackles out-of-distributions as a single set of parameters.
>
> - Thank you for your insightful suggestion. Learning of the coordinate-wise learning rate vector that you suggested may be effective if we can take large number of steps with sufficiently large learning rates. However, in meta-learning scenarios where the model takes only a few gradient steps at each episode, the initialization step becomes significantly more important, and we cannot obtain a similar effect without directly controlling the initialization via $g(z)$. Moreover, adding this balancing variable marginally increases the computational cost since it shares the inference network with other balancing variables. In an earlier version of our model, we had a model that is exactly the same as what you suggested (which learns a task-dependent learning vector), but we empirically found that having separate $\gamma$ and $z$ performs better.
>
> During the rebuttal period, we performed experimental comparison of a modified version of our model (we omitted $\omega$ to see the effect of $\gamma$ and $z$ only), against the model with the task-dependent learning rate vector (denoted as Bayesian Task Dependent $\alpha$) under the same experiment setup we used to obtain the results in Table 1:
>
> Models					                  Omniglot	MNIST
> Bayesian Task Dependent $\alpha$ 	94.19+-0.16	78.40+-0.25
> Bayesian $\gamma$ and $z$ TAML	        96.76+-0.12	81.13+-0.22
>
> The results show that our Bayesian TAML with $\gamma$ and $z$ significantly outperforms the Bayesian MAML with task-dependent learning rate.
>
>
> 3. Experiment on the Meta-Dataset
>
> - Thank you for your helpful suggestion.  We agree, and actually, the OVD dataset in Page 7 is a subset of meta-dataset, on which our model significantly outperforms the baselines (Table 2). While we could not conduct experiments on the full Meta Dataset due to lack of time, we are preparing to conduct this experiment and will include the results on the dataset in the final version of the paper.
>
> 4. Dataset used in Tables 4 and 5.
>
> - The dataset used in Table 4 and 5 is Omniglot. While this is mentioned in the texts, we updated all tables in the Experiment section to clearly describe the dataset in the revision.
>
>
> 5. The meaning of ×5 and ×15 of Table 5 in section 5.2.
>
> - Your interpretation is correct, and this describes the degree of class imbalance. (×N) means that the maximum number of shots is N times larger than the minimum number of shots within the given task. Note that the number of instances for each task is exactly the same to control the task imbalance in this experiment. In Section 5.2, we added more detailed descriptions of the experimental setup.
>
>
> 6. In the Related Work Section, the description for metric-based approaches.
>
> - Thank you for the correction. As you suggested, we revised the descriptions of the  metric-based meta learning approaches in the Related Work section.

---

### Author Response · Authors · 2019-11-15
**Summary of the updates in the revision**

We thank all reviewers for their insightful and constructive comments. Based on the comments, we updated the paper by making the following changes:

* We included the comparison between Bayesian TAML and regularized deterministic TAML suggested by Reviewer #1 (R1), in Table 6.
* We included the dataset names in Table 4 and 5 as suggested by R1.
* We added in more details of the experimental setup for the experiments in Section 5.2, as suggested by R1.
* We revised the discussions on the metric-based meta-learning approaches in the Related Work section, as suggested by R1.
* We added in standard errors to all the results in Section 5.2 and 5.3 as suggested by  Reviewer #3 (R3).
* We reported the values for the three task-specific balancing variables for the complete Bayesian TAML model in Appendix E, as suggested by R3.
* We have corrected the phrase "obtains significantly improves over" to "significantly improves the performance", as suggested by R3.
* We added the experiments for $\theta_0$ initialization in Appendix F, as suggested by Reviewer # 2 (R2).

We believe that our paper became significantly stronger with this revision with all the additional experimental results and corrections, thanks to the suggestions from the reviewers.

---

### Decision · Program_Chairs · 2019-12-19

**Decision:**

Accept (Talk)

**Comment:**

The reviewers generally agreed that the paper presents a compelling method that addresses an important problem. This paper should clearly be accepted, and I would suggest for it to be considered for an oral presentation.

I would encourage the authors to take into account the reviewers' suggestions (many of which were already addressed in the rebuttal period) and my own suggestion.

The main suggestion I would have in regard to improving the paper is to position it a bit more carefully in regard to prior work on Bayesian meta-learning. This is an active research field, with quite a number of papers. There are two that are especially close to the VI method that the authors are proposing: Gordon et al. and Finn et al. (2018). For example, the graphical model in Figure 2 looks nearly identical to the ones presented in these two prior papers, as does the variational inference procedure. There is nothing wrong with that, but it would be appropriate for the authors to discuss this prior work a bit more diligently -- currently the relationship to these prior works is not at all apparent from their discussion in the related work section. A more appropriate way to present this would be to begin Section 3.2 by stating that this framework follows prior work -- there is nothing wrong with building on prior work, and the significant and important contribution of this paper is no way diminished by being up-front about which parts are inspired by previous papers.